# Electrochemotherapy in the Treatment of Bone Metastases: A Systematic Review

**DOI:** 10.3390/jcm12196150

**Published:** 2023-09-23

**Authors:** Maria Beatrice Bocchi, Cesare Meschini, Silvia Pietramala, Andrea Perna, Maria Serena Oliva, Maria Rosaria Matrangolo, Antonio Ziranu, Giulio Maccauro, Raffaele Vitiello

**Affiliations:** 1Department of Orthopaedics, Fondazione Policlinico Universitario A. Gemelli IRCCS, 00168 Rome, Italy; 2Department of Orthopaedics, Università Cattolica del Sacro Cuore, 00168 Rome, Italy; 3Departement of Orthopaedics, Ospedale San Giovanni Evangelista, 00019 Tivoli, Italy

**Keywords:** electrochemotherapy, electroporation, bone metastases

## Abstract

Background and aim: Cancers are one of the most frequent causes of death and disability in humans. Skeletal involvement has a major impact on the quality of life and prognosis of cancer patients. Electrochemotherapy is a palliative and minimally invasive oncologic treatment that was first used to treat subcutaneous nodules for malignant tumors. The aim of our review is to evaluate the results of electrochemotherapy in the treatment of bone metastases. Methods: A systematic review of the literature indexed in the PubMed MEDLINE and Cochrane Library databases using the search key words “electrochemotherapy” AND (“metastasis” OR “metastases”) was performed. The Preferred Reporting Items for Systematic Reviews and MetaAnalyses was followed. Inclusion criteria were proven involvement of the appendicular skeleton in metastatic carcinoma or melanoma, through at least one percutaneous electrochemotherapy session on the metastatic bone lesion. The exclusion criterion was no skeletal metastatic involvement. Results: Eight articles were finally included. We reached a population of 246 patients. The mean age and follow up were 60.1 years old and 11.4 months, respectively. The most represented primary tumor was breast cancer (18.9%). A total of 250 bone lesions were treated with electrochemotherapy. According to RECIST criteria, in our population we observed 55.5% stable diseases. The mean pre-electrochemotherapy VAS value was 6.9, which lowered to 2.7 after treatment. Adverse events occurred in 3.4% of patients. Conclusions: Electrochemotherapy as a minimally invasive and tissue-sparing treatment should be considered for patients with no other alternative to obtain tumor control and improvement in quality of life.

## 1. Introduction

Cancers are one of the most frequent causes of death and disability in humans. An estimated 19 million new cancer diagnoses are made each year. Among the most frequent forms are breast (11.7%), lung (11.4%), colorectal (10%), and prostate (7%) neoplasms [1]. Primary lesions of bone are very rare, representing less than 0.2% of all malignancies. Secondary lesions turn out to be more frequent, most often arising from lung, breast, prostate, kidney, and thyroid cancers; in fact, bone is the third most common site for metastatic disease [2]. Metastases can affect the entire skeleton, with higher incidence in the spine, pelvis, skull, ribs, and long bones such as the humerus and femur [3]. Skeletal involvement has a major impact on the quality of life and prognosis of cancer patients [4]. Patients with metastatic bone disease are evaluated by a team that includes oncologists, radiation oncologists, and orthopedists. The treatment of metastatic lesions depends on various factors including the patient’s life expectancy, general condition, number of metastases, progression of neoplastic disease, sensitivity to treatment, presence of a pathologic fracture, the patient’s quality of life [5,6], and the use of scores such as the Mirel Score [7]. There are various types of treatment for patients with bone metastases. If surgery is needed, several options are available, among which are resection and reconstruction with prosthesis [8], of which custom-made products are increasing in use, or prophylactic synthesis with an intramedullary nail [9]. Chemotherapy or single low-dose radiotherapy can be used as neoadjuvant or adjuvant therapies, with good results but also with higher complication rates [10]. Among pharmacological treatments, the use of inhibitors of osteoclast activity, such as bisphosphonates, and monoclonal therapy, such as denosumab, have had good results [11]. There are also minimally invasive treatments, which include embolization, thermal ablation therapy, high-intensity focused ultrasound, and electrochemotherapy (ECT) [12]. Electrochemotherapy is a palliative minimally invasive oncologic treatment that was first used to treat subcutaneous nodules for malignant tumors in the head and neck region. This technique uses a combination of electroporation and intravenous infusion of chemotherapeutic drugs, such as bleomycin and cisplatin. Electroporation increases the efficacy of drugs by opening cellular transmembrane channels, using permeability-enhancing pulses of electric current, enabling them to reach intracellular targets. The use of electric pulses also induces a transient reduction in tumor blood flow, causing the drug to be trapped in the tissue for a longer time. The cytotoxic action of electrochemotherapy also acts on the endothelial cells of tumor blood vessels, thus reducing the blood supply to the cancer and consequently causing a cascade of tumor cell death surrounding the vessels [13,14]. The main indications for the use of ECT include superficial tumors, superficial metastatic melanoma, breast cancer, head and neck skin tumors, non-melanoma skin cancers, and finally Kaposi sarcoma [15]. With the production of new types of instruments, the type of lesions that can be treated has increased to include deep lesions, although the latter are still being studied and explored [16]. Furthermore, in recent years, many studies on the use of ECT on bone metastases have been published, demonstrating the safety and efficacy of this technique, especially in the field of pain reduction [17] (Figure 1). Unlike other techniques, such as radiotherapy and thermal ablation, ECT has been shown to not affect osteogenic activity or bone integrity [13].

The aim of our review is to evaluate the results of ECT in the treatment of bone metastases.

## 2. Materials and Methods

A systematic review of the literature indexed in the PubMed MEDLINE and Cochrane Library databases using the search key words “electrochemotherapy” AND (“metastasis” OR “metastases”) was performed in August 2023. To minimize the number of missed studies, only the “title/abstract” filter was applied to the search strategy. The bibliography of the selected studies was accurately searched by hand to identify further studies not found during the electronic search. No restrictions on the date of publication or language were applied. The title of the journal, name of authors, and supporting institutions were known at all stages.

The Preferred Reporting Items for Systematic Reviews and MetaAnalyses (PRISMA) was followed as reported in Figure 1. This study was not registered and therefore there is no registration number. This article adhered to the latest Preferred Reporting Items for Systematic Reviews and MetaAnalyses statement [18]. In order to be considered for this review, the articles needed to present some inclusion criteria: proven skeletal involvement in metastatic carcinoma or melanoma and at least one percutaneous ECT session on the metastatic bone lesion.

No skeletal metastatic involvement represented an exclusion criterion.

Abstracts and full texts were independently screened by two authors (R.V. and M.B.B.), and any discordance was solved by consensus with a third author (C.M.). The modified Coleman Methodology Score (mCMS) [19] was used to assess the methodological quality of the studies.

Each article was evaluated by two independent investigators (A.Z. and S.P.); in cases with more than a five-point difference between their ratings, the discrepancy was solved by consensus with a third author (A.P.). The mCMS ranges from 0 to 100 points, representing a well-designed study with no bias or confounding factors. All the selected studies were retrospectively analyzed by an author (M.B.B.) who then extracted and entered the data in an Excel worksheet. The collected data included main author, year of publication, article type, mCMS, patient age and gender, mean follow up, primary tumor, bone involvement, number of procedures, additional previous treatments, local tumor control, clinical outcomes, and potential complications. Local response to therapy was assessed using RECIST (Response Evaluation Criteria in Solid Tumors), which provides a simple and pragmatic methodology to evaluate the activity and efficacy of new cancer therapeutics in solid tumors, in this case for ECT. Pain levels were assessed through the Visual Analogue Scale (VAS), which measures pain intensity. Lastly, the data sheet was reviewed by two authors (R.V. and S.S.) who agreed on the extracted data.

## 3. Results

### 3.1. Articles and Demographic Data

The searches resulted in 294 articles. Following the PRISMA flow chart [18], 16 articles were relevant to the general topic area and finally 8 were included in the review according to inclusion and exclusion criteria (Figure 2) (Table 1) [14,17,20,21,22,23,24,25]. Ranieri G et al. [23] did not exclusively report ECT treatment of skeletal metastases unlike the other authors. It was, however, possible to extrapolate data on their experience in the treating of bone metastases with ECT and thus include this work in our review.

All studies considered were prospective clinical studies except for the works of Cornelis FH et al. [21], Gasbarrini A et al. [22], and Deschamps F et al. [25], which are a case report, a preliminary study, and a retrospective study, respectively. According to the mCMS evaluation, the mean score of the studies reached was 70 points (44–94 points), which represents a suboptimal study design.

We reached a population of 246 patients, of which 105 were males and 141 were females. The mean age of the study population was 60.1 years old. The mean follow up was 11.4 months (Table 1).

### 3.2. Primary Tumor and Bone Metastases Localization

The most-represented primary tumor was breast cancer (18.9%), followed by kidney (16%), lung (10.7%), colon (9.1%), thyroid (6.2%), bladder (4.1%), soft tissue sarcoma (3.3%), endometrium (2.9%), melanoma and prostate (both 2%), and thymoma (1.2%). Finally, in 18.5% and 4.9% of cases the primary tumor was either different from the above or unknown, respectively (Table 2).

A total of 250 bone lesions were treated with ECT. Most of the metastases were in the limbs (54.8%) (upper 20% and lower 80%), while the remaining were located in the trunk (45.2%). Further in detail, the skeletal segments most affected by metastases were the femur (30.1%) and the pelvic ring (26.7%). However, 20.8% of metastatic lesions had costo-vertebral localization instead. More specifically, we are aware that 85% of these were vertebral metastases, since it was specified by some of the authors [22,23,24,26] (Table 3).

### 3.3. Treatments and Outcomes

Many patients underwent previous treatment for bone metastases before ECT: 78 patients received chemotherapy (31.7%), 96 received radiotherapy (39%), 23 received hormone therapy (9.3%), 3 received lesion embolization (1.2%), and finally 55 patients received further unspecified treatments (22.3%).

In all the articles covered by this review, the techniques and methods of performing the ECT procedure were explained in detail. All patients underwent standard X-ray, MRI, or CT scan at presentation to assess tumor volume/density.

The average number of ECT sessions that each individual patient underwent was 1.07 (264/246).

### 3.4. Local Tumor Response

According to RECIST criteria, in our population we observed 12% complete response (CR), 26.1% partial response (PR), 55.5% stable disease (SD), and 6.4% progressive disease (PD) (Table 4).

Ranieri A et al. [23] also assessed local response to therapy with RECIST; however, it was not possible to extrapolate these specific data related to the two patients we considered in the study and included in our review. Only Bianchi G et al. [14] assessed the therapeutic response according to the MD Anderson (MDA) criteria, showing mostly stable disease (85%) and only 10% of the lesions showed progression at follow-up.

Dechamps F et al. [25] and Cornelis FH et al. [21] assessed the local tumor response post ECT on the spine through MRI. While the former also classified the tumor response according to RECIST criteria (Table 4), on the other hand, Cornelis FH et al. [21] merely described the response as a “decreased enhancement” in both patients.

### 3.5. Clinical Outcomes

Pain was assessed by all authors: the mean pre-ECT VAS value was 6.9 (5.1–10), which lowered to 2.7 (1–5) post ECT. A significant decrease in pain intensity and a significant improvement in pharmacologic pain management were observed overall (Table 4).

The health state of patients undergoing ECT was evaluated with the EQ-5D-3L questionnaire by only Campanacci et al. in both the 2021 and 2022 papers [17,20]. The quality-of-life condition improved at follow-up compared with the pre-ECT status. In the latter work, additionally, the impacts of disease on patients’ daily living abilities were evaluated with the ECOG Performance Status Scale [20]. ECOG values 0–1 seemed significantly associated with a higher objective response rate. Likewise, Gasbarrini et al. [22] quantified patients’ functional abilities and the impact of ECT on their basic functional capacities via the Karnofsky Performance Status Scale, reporting a fair improvement of the score post ECT.

### 3.6. Complications

Adverse events occurred in 24 patients (9.7%) and no systemic complications were reported among them (Table 4). Local complications included three long-bone fractures that occurred during the ECT procedure (mostly because of repeated electrode insertion) [14,17,25], three wide skin necroses and ulcerations on previously irradiated areas [14,17,20], and finally one case of neurogenic bladder, which occurred after the third ECT treatment of a large lesion of the sacrum [14].

Peculiar is Deschamps F et al.’s experience [25] of having the highest number of complications (17/40). Complications recorded were acute radicular pain that resolved within a few days (25%), prolonged radicular hypoesthesia (10%), and sub-acute (5%) and acute paraplegia (2.5%). Three months post ECT, 38% of patients had a marked neurological improvement, while 9.5% presented a worsening of neurological symptoms.

## 4. Discussion

To the best of the authors’ knowledge, this is the first systematic review specifically on the use of ECT for the treatment of bone metastases. However, a systematic review of the literature on electrochemotherapy in solid abdominal organ and bone tumors was recently published [26]. The treatment of bone metastases with ECT is discussed but is not the mainstay of the review; moreover, the works cited are a small part of what has been published on the topic in the literature.

Metastatic bone disease is a significant healthcare issue, in fact almost 75% of metastatic bone lesions are symptomatic and require local treatment to improve patients’ outcome and prevent local disease progression [27]. To date, there is no gold-standard treatment for bone metastases. Radiotherapy is the most used local therapy for bone metastases, with estimated rates of pain relief reported in the literature between 50% and 80%. To prove this, all patients treated by Deschamps et al. [25] for instance underwent prior radiotherapy for metastatic epidural spinal cord compression, with, unfortunately, no resolution. This treatment option, however, is not without side effects; therefore, further recent technologies have been used [28,29,30]. The common endpoint to all ablation techniques is the induction of the largest possible thermal necrosis of the target lesion to destroy periosteal nociceptors and reduce tumor size. Compared to other ablation treatments, ECT is minimally invasive, tissue-sparing, and repeatable [31]. Furthermore, ECT ensures good pain control, avoiding negative impacts on quality of life [32].

Primary tumor distribution almost completely overlaps across the literature [1] (breast, kidney, lung, and colon); conversely, the localization of bone metastases does not. According to the data collected, the appendicular skeleton seems to be slightly more involved (54.8%) compared to spine (20.8%) and pelvic (26.7%) localizations. However, this distribution seems to agree with the indications for ECT as an adjuvant therapy in pathological/impending long-bone fractures requiring nailing (femur and tibia) and for lesions that are difficult to access with surgery (pelvis and spine).

In line with data collected in the literature, the radiological response to ECT treatment according to RECIST criteria was mostly a “stable disease” (55.5%); conversely, the number of patients with “disease progression” was negligible (6.4%). In addition, regardless of the classification system used to evaluate the radiological response to ECT (RECIST or MDA [14]), “stable disease” appears to be the most frequent response. In conclusion, local control in bone metastases is achieved with percutaneous ECT.

Pain proved to be a valid clinical outcome indicator, especially in patients who underwent ECT palliative treatments. All the articles included in our review showed a significant reduction in pain symptoms after ECT (VAS values lowered from 6.9 pre ECT to 2.7 post ECT). The decrease appears to be around 60%, showing that ECT obtains results in terms of pain reduction comparable to radiotherapy. Campanacci L et al. [17,20] also observed a reduction in pain even in the absence of an objective radiological response.

ECT does not induce bone necrosis on healthy tissue and does not damage the mineral structure of the bone and its regenerative capacity. Therefore, in case a fracture of the ECT-treated bone occurs, bone healing is possible with usual fracture callus quality and time [33]. In this regard, Campanacci L et al. [17] and Cevolani L et al. [24] studied a slice of the population affected by osteolytic metastasis with either a pathological fracture or an impending fracture, in which there was an indication for preventing nailing fixation. The authors performed tumor electroporation followed by intramedullary nailing: their clinical and radiological results suggested that combining the two procedures does not impact the overall response to the orthopedic treatment. However, Cevolani L et al. [24] suggested that it might be useful to carry out additional local treatments such as radiotherapy or embolization afterward.

Furthermore, since electropermeabilization is a universal phenomenon, any kind of tumor can be treated by ECT, regardless of its usual sensitivity to bleomycin or cisplatin [34].

Regarding the ECT treatment of bone metastases with spinal localization, we found conflicting opinions. Bianchi G et al. [14] considered spine metastases to limit ECT treatment due to the proximity of the spinal cord and technical limitations of electrode positioning. However, it must be said that Bianchi’s group’s experience is certainly the oldest one, and possibly their results and therefore their conclusions may also be influenced by this. In contrast, Gasbarrini A et al. [22], Cornelis FH et al. [21], and not least Deschamps et al. [25] treated spinal metastases only, proving ECT to be a novel and adjuvant treatment of spinal metastases, regardless of the histological types.

Complications related to ECT were few and predictable. The only exception was Deschamps F et al.’s results [25], with the highest number of complications detected. However, it must be considered that this group exclusively examined patients with radiotherapy-resistant metastatic epidural spinal cord compression, which frequently results in neurologic impairment so much so that all patients were assessed pre ECT for any pre-existing neurological symptom. No systemic side effects were described.

Campanacci L et al. [17] have the broadest experience on this matter with the largest series reported in the literature to date. We therefore consider their proposed algorithm on indications for the treatment of bone metastases with ECT to be valid. However, further larger evaluations are mandatory before drawing up a definitive guideline.

Certainly, this literature review cannot be considered to be without limitations. Firstly, it compares data from articles of different types and with different methods of data collection, mainly in terms of outcomes. Furthermore, the lead authors were often co-authors of other papers, with the risk that the populations analyzed may somewhat overlap. However, as it could not be ascertained, we chose not to exclude any article from our review in order to avoid limiting the results and therefore the conclusions.

## 5. Conclusions

Despite the limitations, we can conclude that electrochemotherapy, as a minimally invasive and tissue-sparing treatment, should be considered as an alternative, either alone or with further treatments, for patients with bone metastases to obtain tumor control and improvements in their quality of life.

## Figures and Tables

**Figure 1 jcm-12-06150-f001:**
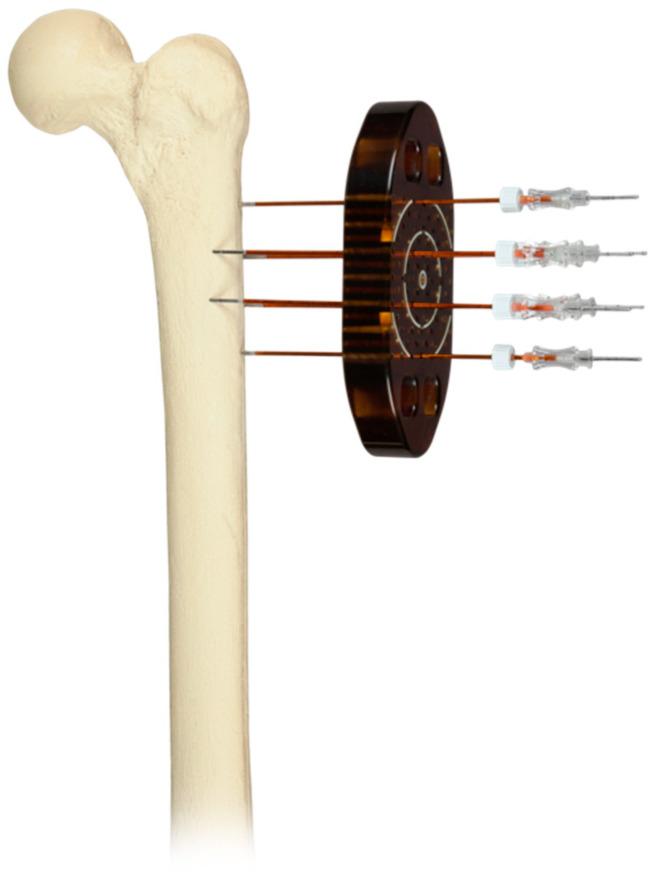
Femur model showing electrode needle placement with a plastic mask used to hold the needles in the correct position during treatment.

**Figure 2 jcm-12-06150-f002:**
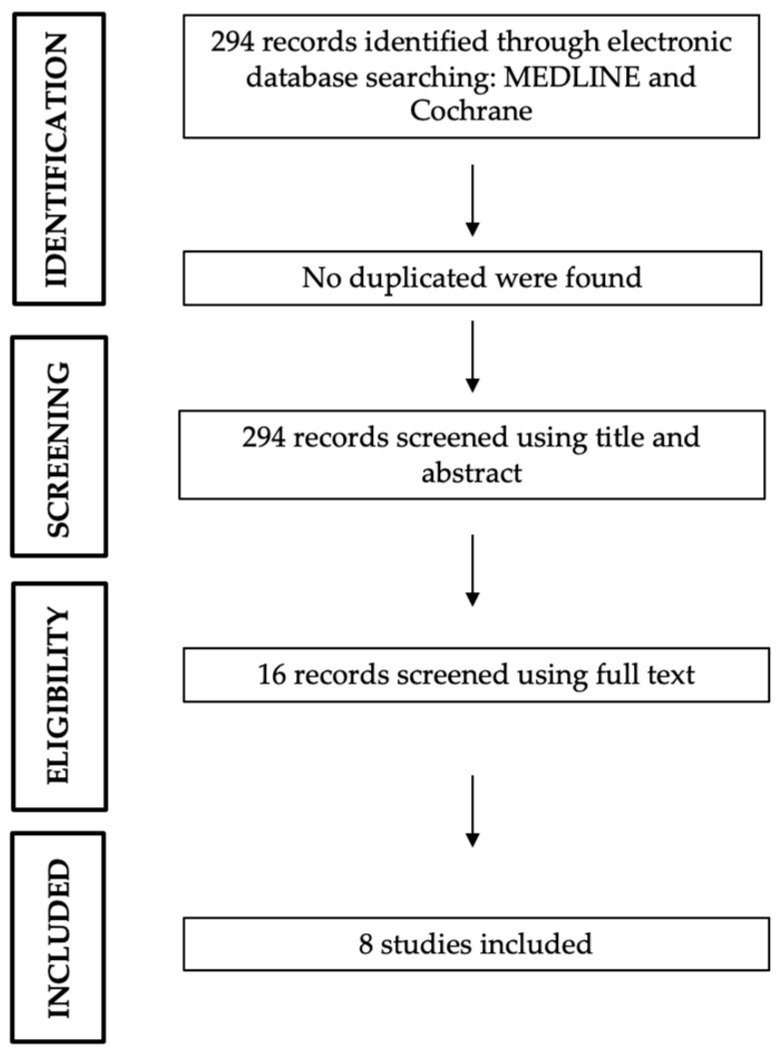
PRISMA2020 flowchart.

**Table 1 jcm-12-06150-t001:** Demographic data.

Refs.	Year of Publication	Manuscript Category	mCMS	GENDER		AGE (Mean)	Follow-Up (Months)
				M	F		
**Bianchi G et al.** [14]	2016	Prospective clinical study	53	10	19	60	7
**Campanacci L et al.** [17]	2021	Prospective clinical study	94	42	60	63	5.9
**Campanacci L et al.** [20]	2022	Prospective clinical study	88	13	25	59	2.2
**Cornelis FH et al.** [21]	2019	Case report	48	1	1	59.5	4
**Gasbarrini A et al.** [22]	2015	Preliminary report	91		1	51	48
**Ranieri G et al.** [23]	2020	Prospective clinical study	44	1	1	65.1	-
**Cevolani L et al.** [24]	2023	Observational prospective study	80	15	17	65	7.7
**Deschamps F et al.** [25]	2023	Retrospective study	70	23	17	58.4	5.1
**Total (mean)**			71	105	141	60.1	11.4

mCMS: modified Coleman Methodology Score.

**Table 2 jcm-12-06150-t002:** Primary tumor localization.

Refs.	Primary Tumor Localization												
	Kidney	Melanoma	Prostate	Breast	Thyroid	Lung	Colon	Thymoma	STS	Bladder	Endometrium	Other	Unk
**Bianchi G et al.** [14]	9	2	1	5	3	1	4		1	1	1		
**Campanacci L et al.** [17]	14		1	23	4	14	10		2	4		21	9
**Campanacci L et al.** [20]	3		1	6		4	4		2	3	6	7	
**Cornelis FH et al.** [21]				1		1							
**Gasbarrini A et al.** [22]		1											
**Ranieri G et al.** [23]				1				1					
**Cevolani L et al.** [24]	10	1	1	9	2	1	2			2		3	1
**Dechamps F et al.** [25]	3	1	1	1	6	5	2	2	3			14	2
**Total**	39	5	5	46	15	26	22	3	8	10	7	45	12

STS, soft-tissue sarcoma; Unk, unknown.

**Table 3 jcm-12-06150-t003:** Bone metastases localization.

Ref.	Bone Metastases Localization											
	Costo/Vertebral	Pelvis	Femur	Fibula	Tibia	Tarsus	Scapula	Ulna	Radius	Humerus	TRUNK	LIMB
**Bianchi G et al.** [14]		12	8		5		1			2	13	15
**Campanacci L et al.** [17]	7	23	33	2	19	3	4	1	2	11	34	71
**Campanacci L et al.** [20]	1	17	6		7	1	3		1	2	21	17
**Cornelis FH et al.** [21]	2										2	
**Gasbarrini A et al.** [22]	1										1	
**Ranieri G et al.** [23]	1	1									2	
**Cevolani L et al.** [24]			18		6					8		32
**Dechamps F et al.** [25]	40										40	
**Total**	52	53	65	2	37	4	8	1	3	23	113	135

**Table 4 jcm-12-06150-t004:** Results.

Refs.	n° Procedure/n° Patients	Pain VAS		RECIST(%)				Complications	
		Pre	Post	CR	PR	SD	PD	Systemic	Local
**Bianchi G et al.** [14]	43/29	6	2.5	-	-	-	-	-	3
**Campanacci L et al.** [17]	105/102	6	2.5	9	16	59	16	-	2
**Campanacci L et al.** [20]	39/38	6	3	2.2	38.2	50.6	9	-	1
**Cornelis FH et al.** [21]	2/2	7.7	5	-	-	-	-	-	-
**Gasbarrini A et al.** [22]	1/1	10	2.75	0	0	100	0	-	-
**Ranieri G et al.** [23]	2/2	7.5	3	-	-	-	-	-	-
**Cevolani L et al.** [24]	32/32	5.1	2.3	3	45	45	7	-	1
**Dechamps F et al.** [25]	40/40	7	1	46	31	23	0	-	17
**Total**	264/246	6.9	2.7	12	26.1	55.5	6.4	-	24

VAS, Visual Analogue Scale; RECIST, Response Evaluation Criteria in Solid Tumors.

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
