# Peer review of "Electrochemotherapy in the Treatment of Bone Metastases: A Systematic Review"

_jcm, 2023, doi:10.3390/jcm12196150_

Round 1

Reviewer 1 Report

Dear Authors, 

There are numerous substantive issues to address:

1. This systematic review included 6 articles relevant to the topic area of “Electrochemotherapy in the Treatment of Bone Metastases”, with a total of 174 patients. In one of the 6 articles (the references Campanacci L et al.19), the “Material and methods” part described “Patients were treated in 11 centers between March 2014 and February 2020. All patients’ data were recorded a shared database (http://reinbone.wng.it)”, and we found in another article of these six articles (the references Campanacci L et al.22), the case data may come from the same database, as described in the paper “2014 saw the start of a registry of patients with bone metastases treated with ECT, whose data are recorded in a shared database. We share the Rizzoli Institute experience of 38 patients treated with ECT for a bone metastasis, excluding patients not included in the registry (before 2014) ”. Besides, we noticed that there are some identical co-authors in these six papers (Campanacci L, Bianchi G, Cornelis FH). So is it possible that these six papers contain a portion of the same case data? How did the authors identify and avoid duplicate case data in these 6 articles? In general, publishing the same data is problematic from the standpoint of systematic review. 

2. Word spelling errorIn the “Abstract” and “Materials and Methods” parts, “elettrochemotherapy” should be corrected as “electrochemotherapy”.

3. Please ensure that every reference cited in the text is also present in the reference list. Please correct the wrong reference numbers cited in all the Tables.

4. A final consideration is that the paper needs extensive English-language editing.

The paper needs extensive English-language editing. I commend the authors for submitting a work like this in a second language but it will require extensive copy editing prior to any publication in English.

Author Response

Question 1.  This systematic review included 6 articles relevant to the topic area of “Electrochemotherapy in the Treatment of Bone Metastases”, with a total of 174 patients. In one of the 6 articles (the references Campanacci L et al.19), the “Material and methods” part described “Patients were treated in 11 centers between March 2014 and February 2020. All patients’ data were recorded a shared database (http://reinbone.wng.it)”, and we found in another article of these six articles (the references Campanacci L et al.22), the case data may come from the same database, as described in the paper “2014 saw the start of a registry of patients with bone metastases treated with ECT, whose data are recorded in a shared database. We share the Rizzoli Institute experience of 38 patients treated with ECT for a bone metastasis, excluding patients not included in the registry (before 2014) ”. Besides, we noticed that there are some identical co-authors in these six papers (Campanacci L, Bianchi G, Cornelis FH). So is it possible that these six papers contain a portion of the same case data? How did the authors identify and avoid duplicate case data in these 6 articles? In general, publishing the same data is problematic from the standpoint of systematic review. 

Authors’ response:

Dear reviewer,

we appreciated your comment. The authors had already posed this same problem at the beginning of the data collection. Unfortunately, it is not possible to separate the different populations and be certain that there is no data overlap. However, we also believe that excluding from our review in advance one or more articles, considering that it is already a niche topic, could have limited the results and thus the conclusions on this matter. Moreover, even if it was the same population partially overlapped, the follow up is longer in more recent articles and this is a thing in terms of outcome assessment. Thanks to your suggestion, however, we thought it useful to make it explicit in the main text that the latter is undoubtedly one of the limitations of our work.

I quote our words in the “Discussion” section:

Certainly, this literature review cannot be considered without limitations. Firstly, it compares data from articles of different types and with different methods of data collection mainly in terms of outcomes. And further, lead authors are often co-authors of other papers with the risk that the populations analyzed may somewhat overlap. However, as it could not be ascertained, we preferred not to exclude any article from our review in order not to risk limiting the results and therefore the conclusions.”.

Question 2.  Word spelling error:In the “Abstract” and “Materials and Methods” parts, “elettrochemotherapy” should be corrected as “electrochemotherapy”.

Authors’ response:

Dear reviewer,

Thank you for your precious remark, we corrected the spelling errors.

Question 3. Please ensure that every reference cited in the text is also present in the reference list. Please correct the wrong reference numbers cited in all the Tables.

Authors’ response:

Dear reviewer,

Thank you for pointing us out these inaccuracies. We have carefully checked the concordance between the main text, the tables and the bibliography.

Question 4.    A final consideration is that the paper needs extensive English-language editing

Authors’ response:

Dear reviewer,

We appreciate your suggestions. A native English speaker has reviewed our paper.

.

Reviewer 2 Report

Authors in their systematic review Electrochemotherapy in the treatment of bone metastasis provide important insight into the existing literature on using electrochemotherapy (i.e. combined treatment of chemotherapy with electroporation) for treatment of bone metastasis. They introduce the importance and difficulties of treating bone metastasis in the introduction. Authors have identified most if not all clinical research papers published until July 2022. Unfortunately they did nor repeat search before finalizing their paper. Namely a paper by Deschamps et al has been published in European Journal of Cancer (https://doi.org/10.1016/j.ejca.2023.03.012) and was available online on 17 March 2023. Deschamps et al. report on treating 40 patients with metastatic epidural spinal cord compression by electrochemotherapy. By including this study additional 40 patients to the reported 174 patients in systematic review would gain considerable weight. In particular as most patinets currently included in the systematic review is from the same institution. I strongly suggest to repeat the search and add new publications including Deschamps et al. Other comments and suggestions are listed below.

Abstract

Authors report that “mean post -ECT VAS value lowered to 3.1”, but they failed to provide pre-ECT VAS value which was 7.2 according to results presented later. As this is one of the most important results I suggest to provide the before/after VAS value already in the abstract.

Materials and Methods

The literature search is adequately described although the search was performed with the combination of search terms “electrochemotherapy” AND “metastasis” in July 2022. Probably more versatile key words could be used. Nevertheless, I believe authors have identified all relevant published clinical study.

Authors present inclusion criteria “proven involvement of appendicular skeleton by metastatic carcinoma or melanoma”. I am curious why also not metastasis in axial skeletal bones are not acceptable; and if authors actually enforced this inclusion criteria – which I have the impression they did not, based on results presented. Authors include Cornelis (ref 22) and Gasbarini (ref 23) which describe treatment of vertebral metastasis only.

Results

It would be interesting if authors could segregate results by sex to investigate if male/female patients respond to ECT differently.

If case reports were not included all studies would come from the same authors/institution so there is considerable danger of bias. Would the results be different if case reports were excluded?

Authors provide distribution according to primary tumor and bone metastasis localization, but do not comment on whether this distribution is representative, i.e. expected.

In describing local tumor response authors include Bianchi (ref 16) whoc assessed the therapeutic response according to MDA (not RECIST criteria) but do not discuss on how these are included/combined.

In Table 4 2nd column (after the authors name and ref, no of procedures are given. In the column there are two number provided (e.g. 43/29). Please provide description as to what means the first and what the second number so that the readers do not need to guess.

The subsection “Features” in the Results section should be better explained. Why are these statements relevant.

Discussion

Authors in the first paragraph state “Radiotherapy is the most used local… with estimated pain relief reported in the literature between 50 and 80%”. Please relate and discuss this number in the context of reduced pain level. How does these 50-80% relate to decrease of VAS score from 7.2 to 3.1?

Better/more relevant references to general description of electrochemotherapy mechanisms of action should be provided (references 30 and 32 are not best suited).

Conclusions

Conclusions should be rewritten. Statement regarding validity of the algorithm proposed by Campanacci et al /ref 19) should be discussed more thoroughly in the Discussion section. In the absence of any discussion it cannot be part of the Conclusions.

Author Response

Question 1. Authors in their systematic review Electrochemotherapy in the treatment of bone metastasis provide important insight into the existing literature on using electrochemotherapy (i.e. combined treatment of chemotherapy with electroporation) for treatment of bone metastasis. They introduce the importance and difficulties of treating bone metastasis in the introduction. Authors have identified most if not all clinical research papers published until July 2022. Unfortunately they did nor repeat search before finalizing their paper. Namely a paper by Deschamps et al has been published in European Journal of Cancer (https://doi.org/10.1016/j.ejca.2023.03.012) and was available online on 17 March 2023. Deschamps et al. report on treating 40 patients with metastatic epidural spinal cord compression by electrochemotherapy. By including this study additional 40 patients to the reported 174 patients in systematic review would gain considerable weight. In particular as most patients currently included in the systematic review is from the same institution. I strongly suggest to repeat the search and add new publications including Deschamps et al.

Authors’ response:

Dear reviewer,

we would particularly like to thank you for this remark. The articles selection for our work was initially done in summer 2022. Since about a year has rightly passed and the literature has been enriched during this time, we thought it more than fair to repeat the research from the beginning. We have thus included the most recent articles from 2023 (references: [24], [25] and [26]).

Question 2. The literature search is adequately described although the search was performed with the combination of search terms “electrochemotherapy” AND “metastasis” in July 2022. Probably more versatile key words could be used. Nevertheless, I believe authors have identified all relevant published clinical study.

Authors’ response:

Dear reviewer,

The main topic of our systematic literature review is very specific and electrochemotherapy is still a niche topic. We believe that using “more versatile key words” would not have served the purpose of our work.

However, since we did the search form the beginning, we modified the search terms as follow: “electrochemotherapy” AND (“metastasis” OR “metastases”).

Question 3. Authors present inclusion criteria “proven involvement of appendicular skeleton by metastatic carcinoma or melanoma”. I am curious why also not metastasis in axial skeletal bones are not acceptable; and if authors actually enforced this inclusion criteria – which I have the impression they did not, based on results presented. Authors include Cornelis (ref 22) and Gasbarini (ref 23) which describe treatment of vertebral metastasis only.

Authors’ response:

Dear reviewer,

Thank you for the correct remark. As you already said, we actually included in our work data from “trunk” metastases (costo-vertebral, pelvis and scapula) and we did several considerations about since the authors experience on this matter is quite different.

I quote our words in the “Materials and Methods” section:

In ordered to be considered for this review the articles needed to present some inclusion criteria: proven skeletal involvement by metastatic carcinoma or melanoma...

Question 4. It would be interesting if authors could segregate results by sex to investigate if male/female patients respond to ECT differently.

Authors’ response:

Dear reviewer,

This is also a very interesting observation, however no authors included this investigation in their work. Also, unfortunately, the way data are presented in all the articles included, has not allowed to segregate the result by gender.

Question 5. If case reports were not included all studies would come from the same authors/institution so there is considerable danger of bias. Would the results be different if case reports were excluded?

Authors’ response:

Dear reviewer,

Even excluding case reports the data would barely change. Considering the final results, data of one or two patients are not significant.

Question 6. Authors provide distribution according to primary tumor and bone metastasis localization, but do not comment on whether this distribution is representative, i.e. expected.

Authors’ response:

Dear reviewer,

We took your precious remark as a cue to enrich or “Discussion”:

“Primary tumor distribution is nearly overlapping with that of literature [1] (breast, kidney, lung and colon), conversely the bone metastases localization does not. According to the data collected, the appendicular skeleton seems to be a little more involved (54.8%) compared to spine (20.8%) and pelvic (26.7%) localizations. However, this distribution seems functional in view of the indications for ECT: on the one hand, adjuvant therapy in pathological/impending long bones fractures requiring nailing (femur, tibia) and on the other hand lesions that are difficult to access with surgery (pelvis, spine).”

Question 7. In describing local tumor response authors include Bianchi (ref 16) who assessed the therapeutic response according to MDA (not RECIST criteria) but do not discuss on how these are included/combined.

Authors’ response:

Dear reviewer,

we specified in the text that our review is not without limitations including heterogeneity of data that made comparison difficult at times. Regarding this specific case, however, it is made explicit in the “Results” section that Ranieri et al used a different system to evaluate the response to therapy. Thanks to your remark, however, we added in the “Discussion” section the following quote to better explain:

“In addition, regardless of the classification system used to evaluate the radiological response to ECT (RECIST or MDA [14]), “stable disease” appears to be the most frequent response. In conclusion, local control in bone metastases is achieved with percutaneous ECT.”

Question 8. In Table 4 2nd column (after the authors name and ref, no of procedures are given. In the column there are two number provided (e.g. 43/29). Please provide description as to what means the first and what the second number so that the readers do not need to guess.

Authors’ response:

Dear reviewer,

To facilitate reading, we added in Table 4 the feature you suggested: n° procedures / n° patients

Question 9. The subsection “Features” in the Results section should be better explained. Why are these statements relevant.

Authors’ response:

Dear reviewer,

In order to be clearer, we delated the “Features” section from the “Results “and we preferred to discuss and compare the peculiarities of the various articles in the “Discussion” section.

Question 10. Authors in the first paragraph state “Radiotherapy is the most used local… with estimated pain relief reported in the literature between 50 and 80%”. Please relate and discuss this number in the context of reduced pain level. How does these 50-80% relate to decrease of VAS score from 7.2 to 3.1?

Authors’ response:

Dear reviewer,

The quote you reported refers to radiotherapy. As much as the reduction in pain we observed with electrochemotherapy is also within this percentage, since we are dealing with data from the literature and that concern very different procedures, we do not think that we can draw comparative conclusions. However, we thought it could be useful to report the data in percentages as you suggested in the “Discussion” section.

“The decrease appears to be around 60% showing that ECT has results on pain reduction comparable to radiotherapy.” 

Question 11. Better/more relevant references to general description of electrochemotherapy mechanisms of action should be provided (references 30 and 32 are not best suited).

Authors’ response:

Dear reviewer,

A more in-depth general description of Electrochemotherapy is provided in the “Introduction” section as well as the relative references.

Question 12.  Conclusions should be rewritten. Statement regarding validity of the algorithm proposed by Campanacci et al /ref 19) should be discussed more thoroughly in the Discussion section. In the absence of any discussion it cannot be part of the Conclusions.

Authors’ response:

Dear reviewer,

We included the assessment related to Campanacci et al’s algorithm in the “Discussion” section as you suggested.

Round 2

Reviewer 1 Report

1. As far as pain relief after ECT is concerned, the outcome might be the result of surgical fixation of the fractured bone. What's you opinion?

2. For readers' better understanding of the technique of ECT, the authors are suggested to give some illustrations or potographs of the ECT procedure in the article.

The quality of English language is good except for a few grammar flows.

Author Response

Dear Editor,

We are pleased to submit our R1 version of the manuscript entitled

Electrochemotherapy in the Treatment of Bone Metastases:  a Systematic Review” 

Manuscript ID: [jcm-2485205]

The reviewers’ comments have been very appreciated. Therefore, an itemized response, structured as a point-by-point answer, is provided below.

________________________________________________________________________________

Reviewer 1

Question 1.  As far as pain relief after ECT is concerned, the outcome might be the result of surgical fixation of the fractured bone. What's your opinion?

Authors’ response:

Dear reviewer,

Thank you for the precious comment.

ECT in combination with surgery has been studied by only a few authors (e.g. Cevolani L et al). The average results presented for what concerns pain are therefore related, for the vast majority, exclusively to treatment with ECT.

Question 2.  For readers' better understanding of the technique of ECT, the authors are suggested to give some illustrations or photographs of the ECT procedure in the article.

Authors’ response:

Dear reviewer,

to make it easier for readers, following your advice, we have enriched our review with a picture that better explore the treatment.

Fig 1. Femur model showing electrode needle placement with a plastic mask used to hold the needles in the correct position during treatment.